# FRÉCHET REGRESSION ON THE BURES-WASSERSTEIN MANIFOLD

**Duc Toan Nguyen**
Department of Electrical
and Computer Engineering
Ken Kennedy Institute
Rice University
Houston, TX, USA
duc.toan.nguyen@rice.edu

**César A. Uribe**
Department of Electrical
and Computer Engineering
Ken Kennedy Institute
Rice University
Houston, TX, USA
cauribe@rice.edu

## ABSTRACT

Fréchet regression, or conditional Barycenters, is a flexible framework for modeling relationships between covariates (usually Euclidean) and response variables on general metric spaces, e.g., probability distributions or positive definite matrices. However, in contrast to classical barycenter problems, computing conditional counterparts in many non-Euclidean spaces remains an open challenge, as they yield non-convex optimization problems with an affine structure. In this work, we study the existence and computation of conditional barycenters, specifically in the space of positive-definite matrices with the Bures-Wasserstein metric. We provide a sufficient condition for the existence of a minimizer of the conditional barycenter problem that characterizes the regression range of extrapolation. Moreover, we further characterize the optimization landscape, proving that under this condition, the objective is free of local maxima. Additionally, we develop a projection-free and provably correct algorithm for the approximate computation of first-order stationary points. Finally, we provide a stochastic reformulation that enables the use of off-the-shelf stochastic Riemannian optimization methods for large-scale setups. Numerical experiments validate the performance of the proposed methods on regression problems of real-world biological networks and on large-scale synthetic Diffusion Tensor Imaging problems.

## 1 INTRODUCTION

Structured prediction on the cone of symmetric positive definite (SPD) matrices arises as diffusion tensors in medical imaging (e.g., DTI) (Pennec et al., 2006; Pennec, 2020), as covariance/precision matrices in statistics and representation learning (Suárez et al., 2021), low-rank matrix recovery problems (Thanwerdas and Pennec, 2023; Maunu et al., 2023), and as graph Laplacians and related operators in network analysis (Zhou and Müller, 2022; Zalles et al., 2024; Calissano et al., 2022; Severn et al., 2021; 2022).

Fréchet regression, also known as *conditional barycenters*, provides a principled way to regress responses valued in a metric space from Euclidean predictors (Petersen and Müller, 2019). Assume the existence of a joint distribution $(X, Y) \sim \mathcal{F}$, where the sample spaces of $X$ and $Y$ are in $(\mathbb{R}^p, \|.\|_2)$ and $(\Omega, d)$, respectively. The Fréchet regression predicts at a query covariate $x$ by solving a *weighted Fréchet mean* problem, i.e., minimizing a weighted sum of squared distances to $Y$, i.e.,

$$
\begin{aligned}
m(x) &= \arg\min_{\omega \in \Omega} \mathbb{E}_{(X,Y) \sim \mathcal{F}} \left[ d^2(Y, \omega) \mid X = x \right] \\
&= \arg\min_{\omega \in \Omega} \mathbb{E}_{(X,Y) \sim \mathcal{F}} \left[ s_G(x) d^2(Y, \omega) \right],
\end{aligned}
\tag{1}
$$

where the weight function is given by $s_G(x) = 1 + (X - \mu)^\top \Sigma^{-1}(x - \mu)$, with $\mu = \mathbb{E}[X]$ and $\Sigma = \text{Var}(X)$. In practice, given $n$ independent samples $(X_k, Y_k) \sim \mathcal{F}$, $k \in \{1, ..., n\}$, the

corresponding empirical estimator of the global function takes the form:

$$\hat{m}_G(x) = \arg\min_{\omega \in \Omega} \frac{1}{n} \sum_{k=1}^{n} s_{G,k}(x) d^2(Y_k, \omega), \tag{2}$$

where $s_{G,k}(x) = 1 + (X_k - \bar{X})^\top \hat{\Sigma}^{-1}(x - \bar{X})$, for $k \in \{1, ..., n\}$, $\bar{X}$ is the sample mean, and $\hat{\Sigma}$ is the sample covariance matrix of $\{X_k\}_{k=1}^n$.

The weights $s_k(x)$ are affine functions of the query $x$, allowing for extrapolation beyond the training covariate range (Figure 1). However, extrapolation inevitably produces **negative weights**, transforming the problem into a *signed barycenter* task. In general, once some $s_{G,k} < 0$, the weighted Fréchet objective can lose coercivity: minimizers may fail to exist in $\Omega$ and minimizing sequences may drift toward the boundary of the space. For SPD-valued responses, this manifests as collapse toward singular positive semidefinite matrices, undermining both the statistical estimator and its computation.

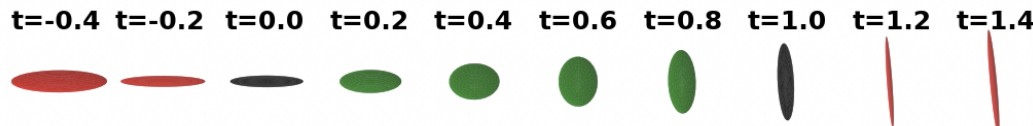

**t=-0.4  t=-0.2  t=0.0  t=0.2  t=0.4  t=0.6  t=0.8  t=1.0  t=1.2  t=1.4**

Figure 1: Interpolation (green) and extrapolation (red) between two (black) ellipsoids (3x3 SPD matrices) under BW metric.

We address this challenge in the Bures-Wasserstein (BW) geometry, a metric preferred for its optimal transport connections and computational efficiency (Bhatia et al., 2019; Han et al., 2021). While BW interpolation (positive weights) is well-understood (Agueh and Carlier, 2011), the signed regime remains an open challenge.

**Contributions.** We propose a robust framework for BW regression with signed weights. First, we establish a **Spectral Dominance** condition that guarantees the existence of the estimator in $\mathbb{S}_{++}^d$, defining a theoretical "safe zone" for extrapolation. Second, we prove that under this condition, Riemannian Gradient Descent is **projection-free**, and we introduce a scalable **Pairwise Stochastic** reformulation for large datasets. Finally, we demonstrate that our method outperforms Frobenius baselines in preserving network topology and scales efficiently to $10^5$ diffusion tensors.

## 2 SPECTRAL DOMINANCE CONDITION FOR THE EXISTENCE OF BURES-WASSERSTEIN BARYCENTERS

The Bures-Wasserstein manifold $\mathbb{S}_{++}^d$ consists of $d \times d$ SPD matrices equipped with the distance arising from optimal transport between zero-mean Gaussian measures. For $S_1, S_2 \in \mathbb{S}_{++}^d$, the squared distance is $W_2^2(S_1, S_2) = \mathrm{Tr}(S_1) + \mathrm{Tr}(S_2) - 2\mathrm{Tr}\left((S_1^{1/2} S_2 S_1^{1/2})^{1/2}\right)$. Additional properties of the Bures-Wasserstein manifold are introduced in Appendix A. Now, we consider the Fréchet regression problem with signed weights on this manifold.

$$\min_{S \in \mathbb{S}_{++}^d} F(S) := \sum_{k=1}^{n} \lambda_k W_2^2(S, \Sigma_k) = \sum_{i \in \mathcal{I}} \lambda_i^+ W_2^2(S, \Sigma_i) - \sum_{j \in \mathcal{J}} \lambda_j^- W_2^2(S, \Sigma_j), \tag{3}$$

for $\lambda_i^+, \lambda_j^- > 0, \mathcal{I} = \{k : \lambda_k > 0\}, \mathcal{J} = \{k : \lambda_k < 0\}, S \in \mathbb{S}_{++}^d$. Without specific constraints, negative weights can drive the solution to the boundary of the cone or to infinity.

**Example 2.1.** *Let* $\lambda_1 = 2, \Sigma_1 = I$ *and* $\lambda_2 = -1, \Sigma_2 = 9I$. *The gradient* $\nabla F(S) = I + S^{-1/2} \succ 0$ *everywhere, implying no minimizer exists in* $\mathbb{S}_{++}^d$.

To ensure well-posedness, we introduce a condition ensuring positive weights "anchor" the solution.

**Theorem 2.2 (Spectral Dominance of Positive Weights).** *Let* $\Sigma_k \in \mathbb{S}_{++}^d$ *with* $\sum_{k=1}^n \lambda_k = 1$. *If*

$$\sum_{i \in \mathcal{I}} \lambda_i^+ \sqrt{\lambda_{\min}(\Sigma_i)} > \sum_{j \in \mathcal{J}} \lambda_j^- \sqrt{\lambda_{\max}(\Sigma_j)}, \tag{4}$$

*then Problem equation 3 admits a solution in $\mathbb{S}_{++}^d$. Furthermore, any stationary point $S_*$ satisfies*

$$\Big(\sum_{i\in\mathcal{I}}\lambda_i^+\sqrt{\lambda_{\min}(\Sigma_i)}-\sum_{j\in\mathcal{J}}\lambda_j^-\sqrt{\lambda_{\max}(\Sigma_j)}\Big)^2 I \prec S_* \prec \Big(\sum_{i\in\mathcal{I}}\lambda_i^+\sqrt{\lambda_{\max}(\Sigma_i)}-\sum_{j\in\mathcal{J}}\lambda_j^-\sqrt{\lambda_{\min}(\Sigma_j)}\Big)^2 I.$$

Condition equation 4 quantifies the dominance of positive mass in the "weakest" directions against negative mass in the "strongest" directions.

**Proposition 2.3.** *Under Theorem 2.2, any stationary point of $F(S)$ is not a local maximum.*

## 3 RIEMANNIAN DESCENT ALGORITHMS FOR CONDITIONAL BURES-WASSERSTEIN BARYCENTER

### 3.1 RIEMANNIAN GRADIENT DESCENT (RGD)

We first consider the full-batch update rule (Algorithm 1).

---
**Algorithm 1** BW Barycenter Gradient Descent (RGD)
---
1: **Update:** $\tilde{S}_t = (1-\eta)I + \eta\sum_{k=1}^n \lambda_k \mathrm{GM}(S_{t-1}^{-1},\Sigma_k)$
2: **Retraction:** $S_t = \tilde{S}_t S_{t-1}\tilde{S}_t$
---

Under the Spectral Dominance condition, we prove that for any $S \in \mathbb{S}_{++}^d$, the updated matrix stays within the SPD cone naturally (Proposition C.1). This **projection-free** property is critical as projecting onto $\mathbb{S}_{++}^d$ costs $O(d^3)$.

**Theorem 3.1** (Convergence of RGD)**.** *Let the conditions of Theorem 2.2 hold, $\eta \leq 1/L$ where $L = \sum_k |\lambda_k|$, $T > 0$ and $S_0 \in \mathbb{S}_{++}^d$. Let $F_*$ be the minimum value of Problem equation 3. Then, the sequence $\{S_t\}_{t=0}^{T-1}$ generated by Algorithm 1 has the following property:*

$$\frac{1}{T}\sum_{t=0}^{T-1}\|\nabla_{\mathrm{bw}}F(S_t)\|_{S_t}^2 \leq \frac{2L(F(S_0)-F_*)}{T}.$$

This sublinear rate is *independent of manifold curvature*. Algorithm 1 exhibits Euclidean-like convergence even on the highly curved Bures-Wasserstein manifold, avoiding the need for compactness assumptions or eigenvalue truncations often required in SPD optimization.

### 3.2 PAIRWISE STOCHASTIC GRADIENT DESCENT (R-SGD)

For large $n$, the full-batch calculation of $n$ geometric means is prohibitive. Standard SGD is risky with negative weights, as sampling a single $\lambda_j < 0$ term could produce an iterate outside the manifold. To resolve this, we propose a **Pairwise Reformulation**:

$$F(S) = \sum_{i\in\mathcal{I},j\in\mathcal{J}}\frac{\lambda_i^+\lambda_j^-}{\mu_+\mu_-}\underbrace{\big(\mu_+ W_2^2(S,\Sigma_i)-\mu_- W_2^2(S,\Sigma_j)\big)}_{f_{ij}(S)},\tag{5}$$

where $\mu_+ := \sum_{i\in\mathcal{I}}\lambda_i^+$ and $\mu_- := \sum_{j\in\mathcal{J}}\lambda_j^-$. Each elementary function $f_{ij}$ is a "mini-barycenter" problem. We apply R-SGD by sampling pairs $(i,j)$.

---
**Algorithm 2** Pairwise Riemannian SGD
---
1: **for** $t = 1,\ldots,T$ **do**
2:      Sample $i \in \mathcal{I}$ w.p. $\lambda_i^+/\mu_+$ and $j \in \mathcal{J}$ w.p. $\lambda_j^-/\mu_-$.
3:      $\tilde{S}_t = (1-\eta_t)I + \eta_t(\mu_+\mathrm{GM}(S_{t-1}^{-1},\Sigma_i)-\mu_-\mathrm{GM}(S_{t-1}^{-1},\Sigma_j))$
4:      $S_t = \tilde{S}_t S_{t-1}\tilde{S}_t$
5: **end for**
---

To guarantee that *every* stochastic step remains in $\mathbb{S}_{++}^d$, we assume a slightly stronger condition:

$$\mu_+ \min_{i \in \mathcal{I}} \sqrt{\lambda_{\min}(\Sigma_i)} > \mu_- \max_{j \in \mathcal{J}} \sqrt{\lambda_{\max}(\Sigma_j)}. \tag{6}$$

Under this condition, Algorithm 2 is fully projection-free. It achieves a convergence rate of $O(\log T/\sqrt{T})$ with diminishing step size Oowada and Iiduka (2025).

## 4 NUMERICAL ANALYSIS

### 4.1 NETWORK REGRESSION ON AN ANT SOCIAL ORGANIZATION

We evaluate our full-batch Algorithm 1 on temporal networks from the Ant Social Organization dataset (Mersch et al., 2013), which is sourced from the Network Repository Rossi and Ahmed (2015). We use the first colony's data (11 days, 113 nodes), representing each daily interaction network as a modified Laplacian $\Sigma := L^\dagger + \frac{1}{d}\mathbf{1}_{d \times d} \in \mathbb{S}_{++}^d$ Haasler and Frossard (2024). The task is to regress the network structure at unobserved time points $X = \tau$ based on the observed sequence. We compare our Bures-Wasserstein (BW) regression against Frobenius (Arithmetic mean) and Fisher-Rao, i.e., Affine-Invariant (Pennec et al., 2006), (Geometric mean) baselines. Figure 2 summarizes the results. Critically, the BW metric excels at preserving topological features. As shown in Figure 2a, our method achieves the lowest Wasserstein distance to the ground truth degree distribution. Furthermore, Figure 2b shows that BW regression preserves the network's modularity (community structure) significantly better than the baselines.

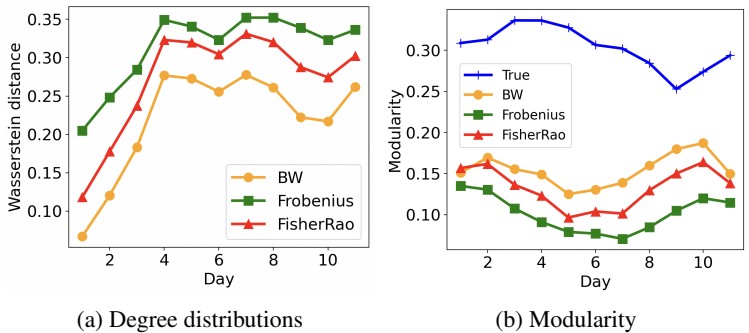

(a) Degree distributions

(b) Modularity

Figure 2: Results on the Ant social organization network dataset

### 4.2 SIMULATED DIFFUSION TENSORS IMAGING (DTI)

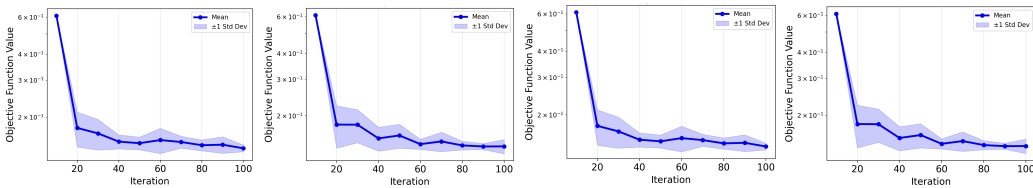

(a) Tensor at index 20000 (b) Tensor at index 40000 (c) Tensor at index 60000 (d) Tensor at index 80000

Figure 3: Objective Values over 100 iterations of Algorithm 2 from regression process of four tensors

We validate the scalability of our Pairwise Riemannian SGD (Algorithm 2) on a synthetic dataset of $n = 100,000$ diffusion tensors generated along a helical trajectory (Fig. 6 in Appendix B). This scale is computationally prohibitive for full-batch methods. We perform Fréchet regression to predict tensors at four equidistant indices (20000, 40000, 60000, 80000) using all other points. Using a diminishing learning rate $\eta_t = \eta_0/\sqrt{t+1}$, Algorithm 2 demonstrates rapid convergence. As shown in Figure 3, across all four target points, there is a sharp initial decline, with the objective value dropping from approximately 0.6 to below 0.2 in just 20 iterations. The variance across 10 independent runs narrows significantly as the step size decays, confirming the stability of the stochastic approximation. Crucially, the pairwise formulation ensures that all $10^5$ iterations remain within the SPD cone $\mathbb{S}_{++}^3$ without requiring expensive projection steps.

## 5    Conclusion

This work resolves the open problem of Bures-Wasserstein Fréchet regression with signed weights. By introducing the Spectral Dominance condition, we guarantee the existence of a minimizer and enable a projection-free algorithmic framework. Our full-batch Riemannian Gradient Descent provably achieves sublinear convergence independent of curvature, while our novel Pairwise R-SGD reformulation effectively scales to datasets of $10^5$ tensors. Empirically, our approach outperforms Euclidean baselines in preserving network topology and demonstrates robust stability in large-scale simulations. Future work will explore accelerated Riemannian methods to speed up convergence and extend this framework to high-dimensional settings via low-rank approximations, further broadening the applicability of BW regression in covariance modeling.

## Acknowledgement

Part of this work is funded by the National Science Foundation under Grant #2443064.

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

## APPENDIX

In this appendix, we first introduce some main property of Bures-Wasserstein manifold. Second, we present detailed visualizations of two experiments from the main paper (section B). Third, we show the proofs of all theorems and propositions precisely in Section C.

## A  PRELIMINARIES ON THE BURES-WASSERSTEIN GEOMETRY

The Bures-Wasserstein manifold $\mathbb{S}^d_{++}$ is the space of $d \times d$ symmetric positive definite (SPD) matrices equipped with the 2-Wasserstein metric arising from optimal transport between zero-mean Gaussian measures. This manifold has a rich geometric structure that enables smooth optimization of covariance matrices and kernel operators. For $S_1, S_2 \in \mathbb{S}^d_{++}$, the squared Bures–Wasserstein distance is

$$W_2^2(S_1, S_2) = \mathrm{Tr}(S_1) + \mathrm{Tr}(S_2) - 2\,\mathrm{Tr}\Big((S_1^{1/2} S_2 S_1^{1/2})^{1/2}\Big).$$

This is precisely the 2-Wasserstein distance between the zero-mean Gaussians $\mathcal{N}(0, S_1)$ and $\mathcal{N}(0, S_2)$. The tangent space at $X \in \mathbb{S}^d_{++}$, denoted by $T_X \mathbb{S}^d_{++}$, is the linear space $\mathrm{Sym}(d)$ of symmetric matrices. The Bures-Wasserstein Riemannian metric is defined using the Lyapunov operator, $\mathcal{L}_X[U]$, which is the solution of $XZ + ZX = U$. For $U, V \in T_X \mathbb{S}^d_{++}$, the Riemannian metric is $g_{\mathrm{bw}}(U, V) = \frac{1}{2} \mathrm{Tr}(\mathcal{L}_X[U]\,V)$. The Riemannian exponential map is $\mathrm{Exp}_{\mathrm{bw}, X}(U) = X + U + \mathcal{L}_X[U]\,X\,\mathcal{L}_X[U]$, for $U \in T_X \mathbb{S}^d_{++}$. Given a smooth function $f : \mathbb{S}^d_{++} \to \mathbb{R}$, with Euclidean gradient $\nabla f(X)$, the Bures-Wasserstein gradient is $\nabla_{\mathrm{bw}} f(X) = 4 \{\nabla f(X)\, X\}_{\mathrm{s}}$, where $\{\cdot\}_{\mathrm{s}}$ denotes the symmetrization operator $\{A\}_{\mathrm{s}} = \frac{1}{2}(A + A^\top)$. The manifold $\mathbb{S}^d_{++}$ is geodesically convex and non-negatively curved. Between any $S_1, S_2 \in \mathbb{S}^d_{++}$, the unique constant-speed geodesic is $\gamma(t) = \big((1-t)I + t\,T\big) S_1 \big((1-t)I + t\,T\big)$, for $t \in [0, 1]$, where $T = S_1^{-1/2}(S_1^{1/2} S_2 S_1^{1/2})^{1/2} S_1^{-1/2}$. The term $T$ is also equivalent to the geometric mean of $S_1^{-1}$ and $S_2$. We denote the geometric mean between two SPD matrices $A, B$ as $\mathrm{GM}(A, B) = A^{1/2}(A^{-1/2} B A^{-1/2})^{1/2} A^{1/2}$. For each matrix $S \in \mathbb{S}^d_{++}$, we denote the minimum and maximum eigenvalues of $S$ as $\lambda_{\min}(S)$ and $\lambda_{\max}(S)$, respectively.

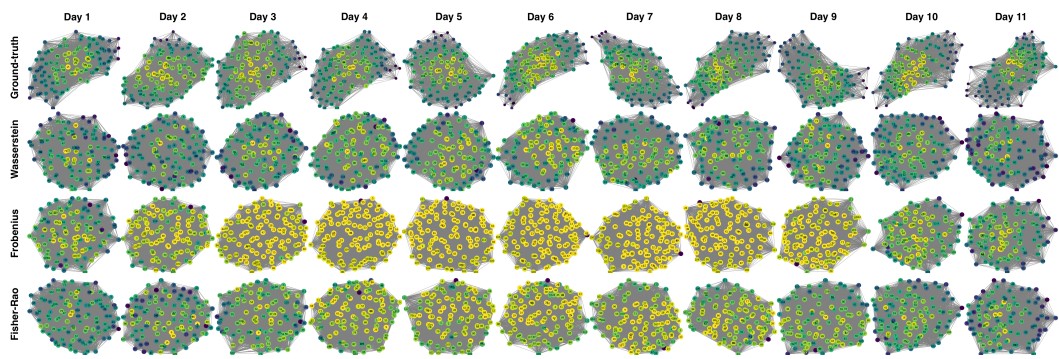

Figure 4: Network plots for ground truth and three methods on 11 days

# B    ADDITIONAL VISUALIZATION

## B.1    ANT SOCIAL ORGANIZATION NETWORK

Figure 4 illustrates the networks generated by the Wasserstein, Frobenius, and Fisher-Rao regression methods compared against the ground truth. We use a spring-force layout in which each node's color corresponds to its degree.

As observed in the plots, the Frobenius and Fisher-Rao methods, particularly during the interpolation range (i.e., Days 4 to 8), exhibit a "smoothing" effect. In these rows, the node colors become largely uniform, indicating that the degree distribution has homogenized and the distinct structural features are lost. In contrast, our proposed Wasserstein method preserves the heterogeneity of the degrees throughout the timeline. The color variation in the Wasserstein row remains distinct and closely mirrors the diverse degree distribution seen in the ground truth networks.

## B.2    SIMPLE FRÉCHET REGRESSION EXAMPLE WITH DIFFUSION TENSORS

This figure provides a minimal illustration of Fréchet regression applied to diffusion tensors, modeled as $3 \times 3$ SPD matrices under the Bures–Wasserstein metric. Given two observed diffusion tensors (black), the fitted regression curve corresponds to the unique geodesic connecting them. As the covariate parameter $t$ varies within the observed range $[0, 1]$, the predicted tensors (green) smoothly interpolate between the endpoints, preserving positive definiteness and yielding physically meaningful diffusion ellipsoids. Conversely, when $t$ extends outside this range, the regression produces extrapolated tensors (red) that rapidly become highly anisotropic (flattened) or nearly degenerate.

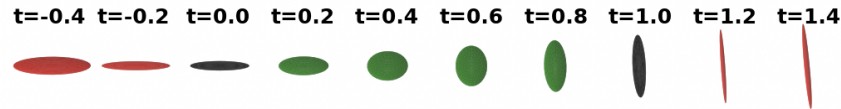

Figure 5: Interpolation (green) and extrapolation (red) between two (black) ellipsoids (3x3 SPD matrices) under the BW metric

## B.3    HELIX TENSORS PREDICTION (DTI)

We visualize the regression results along the helical trajectory in Figure 6. Figure 6a displays 20 representative ground-truth tensors (subsampled from the $100,000$ dataset) along the helical backbone, visualized as ellipsoids. Figure 6b shows the corresponding predicted tensors resulting from our Fréchet regression model. The color of each ellipsoid in Figure 6b encodes the Bures-Wasserstein error magnitude relative to the ground truth. Visually, the predicted tensors closely match the orientation and anisotropy of the ground truth. This is quantitatively confirmed by the color scale; the

majority of the tensors are dark blue, indicating small Bures-Wasserstein distances near 0.05. A notable exception is the yellow ellipsoid at the base of the helix (near $z = 0$). This point represents the furthest extrapolation boundary in this set, resulting in a higher error magnitude of approximately 0.35. Overall, these visualizations demonstrate that our R-SGD optimizer effectively recovers diffusion tensors even along highly non-linear spatial curves, indicating strong potential for real-world DTI applications.

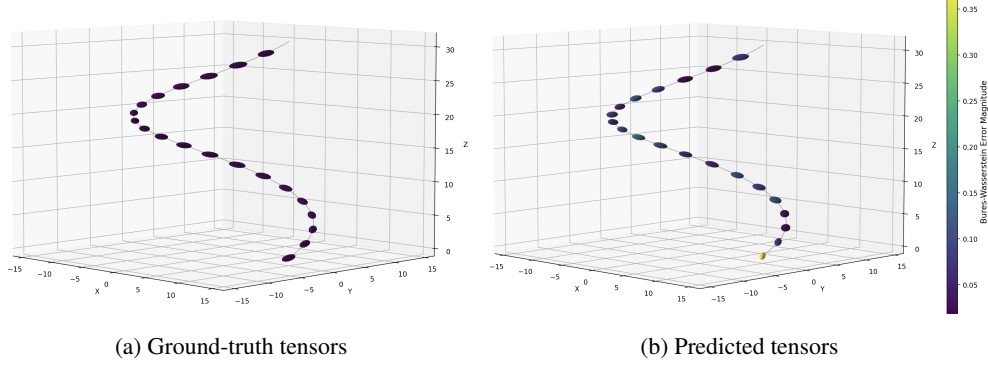

(a) Ground-truth tensors                             (b) Predicted tensors

Figure 6: Helix visualization for 20 ground-truth tensors (from 100,000 samples) and their prediction from the model

## C    PROOFS FOR MAIN THEOREMS AND PROPOSITIONS

In this appendix section, we will show in detail the proofs for Theorem 2.2, Proposition 2.3, and Theorem 3.1. First, we come to the proof of our main theorem.

***Proof for Theorem 2.2.*** First, we use the Löwner-Heinz theorem (Löwner, 1934), which states that the function $f(t) = t^{1/2}$ is operator monotone on $(0, \infty)$. That is, for $A, B \succ 0$, if $A \preceq B$, then $A^{1/2} \preceq B^{1/2}$. We also use the nuclear norm identity

$$\mathrm{Tr}\big((S^{1/2}\Sigma S^{1/2})^{1/2}\big) \;=\; \big\|\Sigma^{1/2} S^{1/2}\big\|_*, \tag{7}$$

which follows from $\|A\|_* = \mathrm{Tr}((A^\top A)^{1/2})$ applied to $A = \Sigma^{1/2} S^{1/2}$. Let $\mathbb{S}^d_+ = \{\Sigma \in \mathbb{R}^{d \times d} : \Sigma^\top = \Sigma \text{ and } \Sigma \succeq 0\}$ is the set of positive semi-definite matrices. For convenience, we use some notations:

$$A_{\min} = \sum_{i \in \mathcal{I}} \lambda_i^+ \sqrt{\lambda_{\min}(\Sigma_i)}, \quad B_{\min} = \sum_{j \in \mathcal{J}} \lambda_j^- \sqrt{\lambda_{\min}(\Sigma_j)},$$

$$A_{\max} = \sum_{i \in \mathcal{I}} \lambda_i^+ \sqrt{\lambda_{\max}(\Sigma_i)}, \quad B_{\max} = \sum_{j \in \mathcal{J}} \lambda_j^- \sqrt{\lambda_{\max}(\Sigma_j)}.$$

We proceed in two steps.

**Step 1: Coercivity and existence on the closed cone $\mathbb{S}^d_+$.** For each $k \in \{1, ..., n\}$, from $\lambda_{\min}(\Sigma_k) I \preceq \Sigma_k \preceq \lambda_{\max}(\Sigma_k) I$ and operator monotonicity,

$$\sqrt{\lambda_{\min}(\Sigma_k)}\, S^{1/2} \;\preceq\; (S^{1/2}\Sigma_k S^{1/2})^{1/2} \;\preceq\; \sqrt{\lambda_{\max}(\Sigma_k)}\, S^{1/2}.$$

Taking traces and separating positive/negative weights gives

$$\sum_{k=1}^n \lambda_k \, \mathrm{Tr}\big((S^{1/2}\Sigma_k S^{1/2})^{1/2}\big) \;\leq\; \left(\sum_{i \in \mathcal{I}} \lambda_i^+ \sqrt{\lambda_{\max}(\Sigma_i)} - \sum_{j \in \mathcal{J}} \lambda_j^- \sqrt{\lambda_{\min}(\Sigma_j)}\right) \mathrm{Tr}(S^{1/2})$$

$$= (A_{\max} - B_{\min})\mathrm{Tr}(S^{1/2}).$$

Therefore,

$$F(S) \geq \mathrm{Tr}S - 2(A_{\max} - B_{\min})\,\mathrm{Tr}(S^{1/2}) + \sum_{k=1}^{n} \lambda_k \mathrm{Tr}\Sigma_k. \tag{8}$$

Let $\lambda_1', \ldots, \lambda_d'$ be the eigenvalues of $S$. By the Cauchy-Schwarz inequality,

$$\mathrm{Tr}(S^{1/2}) = \sum_{l=1}^{d} \sqrt{\lambda_l'} \leq \sqrt{d \sum_{l=1}^{d} \lambda_l'} = \sqrt{d\,\mathrm{Tr}S}.$$

Setting $x := \mathrm{Tr}S$ and $c_1 := A_{\max} - B_{\min}$, the right–hand side of equation 8 is

$$x - 2c_1\sqrt{dx} + \mathrm{const} = \left(\sqrt{x} - c_1\sqrt{d}\right)^2 - c_1^2 d + \mathrm{const} \xrightarrow[x\to\infty]{} +\infty.$$

Hence $F$ is *coercive* on the closed cone $\mathbb{S}_+^d$. Because $F$ is continuous, it attains a minimum on $\mathbb{S}_+^d$ (Weierstrass on closed, bounded sublevel sets). Denote one such minimizer by $\widehat{S} \succeq 0$.

**Step 2: Under equation 4, no singular matrix minimizes $F$.** Assume, for contradiction, that the minimizer $\widehat{S}$ is singular. Let $m := \dim \ker \widehat{S} \geq 1$, and let $U \in \mathbb{R}^{d \times m}$ have orthonormal columns spanning $\ker \widehat{S}$. Define the orthogonal projector $P_{\mathrm{ker}} := UU^\top$ and, for $t > 0$, set

$$S(t) := \widehat{S} + t\,P_{\mathrm{ker}} \in \mathbb{S}_{++}^d.$$

In an orthonormal basis in which the first $m$ coordinates span $\ker \widehat{S}$, we can write

$$\widehat{S} = \mathrm{diag}(0_m,\, \widehat{S}_\perp), \qquad S(t) = \mathrm{diag}(tI_m,\, \widehat{S}_\perp),$$

so that

$$S(t)^{1/2} = \mathrm{diag}(\sqrt{t}\,I_m,\, \widehat{S}_\perp^{1/2}), \qquad S(t)^{1/2} - \widehat{S}^{1/2} = \mathrm{diag}(\sqrt{t}\,I_m,\, 0), \tag{9}$$

and consequently

$$\left\| S(t)^{1/2} - \widehat{S}^{1/2} \right\|_* = m\sqrt{t}, \qquad U^\top S(t)^{1/2}U = \sqrt{t}\,I_m, \qquad U^\top \widehat{S}^{1/2}U = 0. \tag{10}$$

We define

$$\Delta F(t) = F(S(t)) - F(\widehat{S})$$

$$= t\,\mathrm{Tr}(P_{\mathrm{ker}}) - 2\sum_{k=1}^{n} \lambda_k\,\Delta_k(t)$$

$$= tm - 2\sum_{k=1}^{n} \lambda_k\,\Delta_k(t),$$

where

$$\Delta_k(t) := \mathrm{Tr}\big((S(t)^{1/2}\Sigma_k S(t)^{1/2})^{1/2}\big) - \mathrm{Tr}\big((\widehat{S}^{1/2}\Sigma_k \widehat{S}^{1/2})^{1/2}\big).$$

We now bound $\Delta_k(t)$ for positive and negative weights.

- *Upper bound (used when $\lambda_k < 0$).* Using equation 7 and the triangle inequality for the nuclear norm,

$$\Delta_k(t) = \left\|\Sigma_k^{1/2}S(t)^{1/2}\right\|_* - \left\|\Sigma_k^{1/2}\widehat{S}^{1/2}\right\|_*$$

$$\leq \left\|\Sigma_k^{1/2}\big(S(t)^{1/2} - \widehat{S}^{1/2}\big)\right\|_*$$

$$\leq \|\Sigma_k^{1/2}\|_{\mathrm{op}} \left\|S(t)^{1/2} - \widehat{S}^{1/2}\right\|_*.$$

Since $\|\Sigma_k^{1/2}\|_{\mathrm{op}} = \sqrt{\lambda_{\max}(\Sigma_k)}$ and by equation 10,

$$\Delta_k(t) \leq \sqrt{\lambda_{\max}(\Sigma_k)}\,m\sqrt{t}. \tag{11}$$

- *Lower bound (used when $\lambda_k > 0$).* Let $A_k(t) := (S(t)^{1/2}\Sigma_k S(t)^{1/2})^{1/2}$. Because $A_k(0) \succeq 0$ and $A_k(0)U = 0$ (since $\widehat{S}^{1/2}U = 0$), we have

$$\Delta_k(t) \geq \mathrm{Tr}(U^\top A_k(t)U).$$

Moreover, $\Sigma_k \succeq \lambda_{\min}(\Sigma_k)I$ implies $A_k(t) \succeq \sqrt{\lambda_{\min}(\Sigma_k)}\, S(t)^{1/2}$ by operator monotonicity. Hence, using equation 10,

$$\Delta_k(t) \;\geq\; \sqrt{\lambda_{\min}(\Sigma_k)}\,\mathrm{Tr}(U^\top S(t)^{1/2}U) \;=\; \sqrt{\lambda_{\min}(\Sigma_k)}\, m\sqrt{t}. \tag{12}$$

Splitting the sum according to the sign of $\lambda_k$ and combining equation 11–equation 12, we obtain

$$\sum_{k=1}^n \lambda_k \, \Delta_k(t) \;\geq\; m\left( \sum_{i\in\mathcal{I}} \lambda_i^+ \sqrt{\lambda_{\min}(\Sigma_i)} > \sum_{j\in\mathcal{J}} \lambda_j^- \sqrt{\lambda_{\max}(\Sigma_j)} \right)\sqrt{t}$$
$$=\; m(A_{\min} - B_{\max})\sqrt{t}$$

Therefore

$$\Delta F(t) \;\leq\; tm - 2m(A_{\min} - B_{\max})\sqrt{t} \;=\; m\Big(t - 2(A_{\min} - B_{\max})\sqrt{t}\Big).$$

Let $c_2 := A_{\min} - B_{\max} > 0$ under equation 4. For every $t \in (0, c_2^2)$,

$$\Delta F(t) \;\leq\; m(\sqrt{t} - c_2)^2 - mc_2^2 < 0.$$

Thus $F(S(t)) < F(\widehat{S})$ for all sufficiently small $t > 0$, contradicting the minimality of the singular $\widehat{S}$ on $\mathbb{S}_+^d$.

Since a minimizer on $\mathbb{S}_+^d$ exists (Step 1) and no singular matrix can be optimal under equation 4 (Step 2), the minimum is attained at some $S^\star \in \mathbb{S}_{++}^d$. $\qquad\square$

Next, we further show that under our existence condition, there is no local maximum in the domain $\mathbb{S}_{++}^d$.

***Proof for Proposition 2.3.*** Since the function $F(S)$ is differentiable on $\mathbb{S}_{++}^d$, under condition (4), there is at least one stationary point $S_*$ that satisfies

$$S_* = \sum_k \lambda_k \left( S_*^{1/2}\Sigma_k S_*^{1/2} \right)^{1/2}. \tag{13}$$

Let $1 > \varepsilon > 0$. Consider $S_t := tS_*$, for $t \in (1-\varepsilon, 1+\varepsilon)$, such that $S_t \in \mathbb{S}_{++}^d$ for all $t$. We have

$$F(S_t) = \sum_{k=1}^n \lambda_k W_2^2(S_t, \Sigma_k)$$
$$= \sum_{k=1}^n \lambda_k \left( \mathrm{Tr}(\Sigma_k) + \mathrm{Tr}(S_t) - 2\mathrm{Tr}\left( S_t^{1/2}\Sigma_k S_t^{1/2} \right)^{1/2} \right)$$
$$= \sum_{k=1}^n \lambda_k \mathrm{Tr}(\Sigma_k) + \mathrm{Tr}(S_t) - 2\mathrm{Tr}\left( \sum_k \lambda_k \left( S_t^{1/2}\Sigma_k S_t^{1/2} \right)^{1/2} \right)$$
$$= \sum_{k=1}^n \lambda_k \mathrm{Tr}(\Sigma_k) + t\,\mathrm{Tr}(S_*) - 2\sqrt{t}\,\mathrm{Tr}\left( \sum_k \lambda_k \left( S_*^{1/2}\Sigma_k S_*^{1/2} \right)^{1/2} \right)$$
$$= \sum_{k=1}^n \lambda_k \mathrm{Tr}(\Sigma_k) + t\,\mathrm{Tr}(S_*) - 2\sqrt{t}\,\mathrm{Tr}(S_*) \quad \text{(From equation (13))}$$
$$= \sum_{k=1}^n \lambda_k \mathrm{Tr}(\Sigma_k) + (t - 2\sqrt{t})\,\mathrm{Tr}(S_*).$$

Denote $f(t) := t - 2\sqrt{t}$ on $(1 - \varepsilon, 1 + \varepsilon)$. We have

$$f'(t) = \frac{\sqrt{t} - 1}{\sqrt{t}}.$$

From that, $f'(1) = 0$, $f'(t) < 0$ on $(1 - \varepsilon, 1)$, and $f'(t) > 0$ on $(1, 1 + \varepsilon)$. Thus, $f(t)$ has a local minimum at $t = 1$ on $(1 - \varepsilon, 1 + \varepsilon)$. Since $\text{Tr}(S_*) > 0$, then $F$ also has a local minimum at $S_*$ on the smooth curve $S_t$ for $t \in (1 - \varepsilon, 1 + \varepsilon)$. Therefore, $S_*$ cannot be a local maximum. $\square$

Now, we prove that all iterations of Algorithm 1 stay inside the domain under our condition.

**Proposition C.1.** *Let $\Sigma_k \in \mathbb{S}_{++}^d$ and corresponding weights $\lambda_k \in \mathbb{R}$, for $k \in \{1, ..., n\}$, such that the Spectral Dominance of Positive Weights in Theorem 2.2 holds. Then, for any $S \in \mathbb{S}_{++}^d$, $\sum_{k=1}^n \lambda_k \text{GM}(S^{-1}, \Sigma_k) \in \mathbb{S}_{++}^d$.*

*Proof.* Let $\Sigma_k \in \mathbb{S}_{++}^d$, $\lambda_k \in \mathbb{R}$, for $k \in \{1, ..., n\}$, satisfying condition (4). For any $S \in \mathbb{S}_{++}^d$,

$$\sum_{k=1}^n \lambda_k \text{GM}(S^{-1}, \Sigma_k) = \sum_{i \in \mathcal{I}} \lambda_i^+ \text{GM}(S^{-1}, \Sigma_i) - \sum_{j \in \mathcal{J}} \lambda_j^- \text{GM}(S^{-1}, \Sigma_j)$$

$$= \sum_{i \in \mathcal{I}} \lambda_i^+ S^{-1/2} (S^{1/2} \Sigma_i S^{1/2})^{1/2} S^{-1/2} - \sum_{j \in \mathcal{J}} \lambda_j^- S^{-1/2} (S^{1/2} \Sigma_j S^{1/2})^{1/2} S^{-1/2}$$

$$= S^{-1/2} \left( \sum_{i \in \mathcal{I}} \lambda_i^+ (S^{1/2} \Sigma_i S^{1/2})^{1/2} - \sum_{j \in \mathcal{J}} \lambda_j^- (S^{1/2} \Sigma_j S^{1/2})^{1/2} \right) S^{-1/2}$$

Denote $M := \sum_{i \in \mathcal{I}} \lambda_i^+ (S^{1/2} \Sigma_i S^{1/2})^{1/2} - \sum_{j \in \mathcal{J}} \lambda_j^- (S^{1/2} \Sigma_j S^{1/2})^{1/2}$. We will show that $M \succ 0$.

Similar to the proof of Theorem 2.2, for $A, B \succ 0$, if $A \preceq B$, then $A^{1/2} \preceq B^{1/2}$.

First, consider the positive weight terms. Since $\Sigma_i \succeq \lambda_{\min}(\Sigma_i) I$, conjugating by $S^{1/2}$ gives $S^{1/2} \Sigma_i S^{1/2} \succeq \lambda_{\min}(\Sigma_i) S$. Applying the Löwner-Heinz theorem:

$$(S^{1/2} \Sigma_i S^{1/2})^{1/2} \succeq \sqrt{\lambda_{\min}(\Sigma_i)} S^{1/2}.$$

Similarly, for the negative weight terms, we have $\Sigma_j \preceq \lambda_{\max}(\Sigma_j) I$, which implies $S^{1/2} \Sigma_j S^{1/2} \preceq \lambda_{\max}(\Sigma_j) S$. Applying the theorem again:

$$(S^{1/2} \Sigma_j S^{1/2})^{1/2} \preceq \sqrt{\lambda_{\max}(\Sigma_j)} S^{1/2}.$$

Substituting these bounds back into the expression for $M$:

$$M = \sum_{i \in \mathcal{I}} \lambda_i^+ (S^{1/2} \Sigma_i S^{1/2})^{1/2} - \sum_{j \in \mathcal{J}} \lambda_j^- (S^{1/2} \Sigma_j S^{1/2})^{1/2}$$

$$\succeq \sum_{i \in \mathcal{I}} \lambda_i^+ \left( \sqrt{\lambda_{\min}(\Sigma_i)} S^{1/2} \right) - \sum_{j \in \mathcal{J}} \lambda_j^- \left( \sqrt{\lambda_{\max}(\Sigma_j)} S^{1/2} \right)$$

$$= \left( \sum_{i \in \mathcal{I}} \lambda_i^+ \sqrt{\lambda_{\min}(\Sigma_i)} - \sum_{j \in \mathcal{J}} \lambda_j^- \sqrt{\lambda_{\max}(\Sigma_j)} \right) S^{1/2}.$$

Let $\Delta := \sum_{i \in \mathcal{I}} \lambda_i^+ \sqrt{\lambda_{\min}(\Sigma_i)} - \sum_{j \in \mathcal{J}} \lambda_j^- \sqrt{\lambda_{\max}(\Sigma_j)}$. By the hypothesis, we have $\Delta > 0$. Since $S \in \mathbb{S}_{++}^d$, its square root $S^{1/2}$ is also strictly positive definite. Therefore, $M \succeq \Delta S^{1/2} \succ 0$.

Finally, since $M \succ 0$ and $S^{-1/2}$ is invertible, the congruence transformation $S^{-1/2} M S^{-1/2}$ preserves positive definiteness. Thus,

$$\sum_{k=1}^n \lambda_k \text{GM}(S^{-1}, \Sigma_k) \in \mathbb{S}_{++}^d.$$

$\square$

Finally, we prove the sublinear convergence rate of Algorithm 1. First, recall the $1-$smoothness of the Bures-Wasserstein distance function (Chewi et al., 2020, Theorem 7). This property is useful for the following auxiliary results.

**Lemma C.2.** *Let* $X, Y, \Sigma \in \mathbb{S}^d_{++}$. *Define* $G(X) := W_2^2(X, \Sigma)$. *Then,* $G(Y) \leq G(X) + \langle \nabla_{\mathrm{bw}} G(X), \log_X Y \rangle_X + \frac{1}{2} W_2^2(X, Y)$. *This inequality is equivalent to,*

$$\|\nabla_{\mathrm{bw}} G(Y) - \Gamma_X^Y(\nabla_{\mathrm{bw}} G(X))\|_Y \leq W_2(X, Y), \tag{14}$$

*where* $\Gamma_X^Y$ *is the parallel transport from* $T_X \mathbb{S}^d_{++}$ *to* $T_Y \mathbb{S}^d_{++}$ *along the geodesic* $\gamma$ *connecting* $X$ *and* $Y$ *with* $\gamma(0) = X$, $\gamma(1) = Y$.

**Lemma C.3.** *From (14), we have the function* $-G(X)$ *is also* $1-$*smooth, and thus* $-G(Y) \leq -G(X) + \langle -\nabla_{\mathrm{bw}} G(X), \log_X Y \rangle_X + \frac{1}{2} W_2^2(X, Y)$.

From these properties, we imply the $L-$smoothness of the objective function

**Lemma C.4.** *The function* $F(S)$ *is* $L-$*smooth with* $L = \sum_k |\lambda_k| > 1$.

Lemma C.4 allows us to prove the convergence rate for Algorithm 1 as follows.

***Proof for Theorem 3.1.*** At each iteration $t + 1$, $S_{t+1} = \exp_{S_t}(-\eta \nabla_{\mathrm{bw}} F(S_t))$. By the $L-$smoothness of the objective function, we have

$$F(S_{t+1}) \leq F(S_t) + \langle \nabla_{\mathrm{bw}} F(S_t), \log_{S_t}(S_{t+1}) \rangle_{S_t} + \frac{L}{2} W_2^2(S_t, S_{t+1})$$

$$= F(S_t) + \langle \nabla_{\mathrm{bw}} F(S_t), -\eta \nabla_{\mathrm{bw}} F(S_t) \rangle_{S_t} + \frac{L}{2} \eta^2 \|\nabla_{\mathrm{bw}} F(S_t)\|_{S_t}^2$$

$$= F(S_t) - \eta \left(1 - \frac{\eta L}{2}\right) \|\nabla_{\mathrm{bw}} F(S_t)\|_{S_t}^2$$

$$\leq F(S_t) - \frac{\eta}{2} \|\nabla_{\mathrm{bw}} F(S_t)\|_{S_t}^2 \quad \text{(Since } \eta < 1/L\text{).}$$

From this inequality, we have that $F(S)$ decreases after every iteration. Also, if we choose $\eta = 1/L$, then by applying the inequality for multiple iterations, we have

$$F(S_T) \leq F(S_{T-1}) - \frac{1}{2L} \|\nabla_{\mathrm{bw}} F(S_{T-1})\|_{S_{T-1}}^2$$

$$\leq F(S_{T-2}) - \frac{1}{2L} \left(\|\nabla_{\mathrm{bw}} F(S_{T-1})\|_{S_{T-1}}^2 + \|\nabla_{\mathrm{bw}} F(S_{T-2})\|_{S_{T-2}}^2\right)$$

Continue this iterated inequality, we have

$$F(S_T) \leq F(S_0) - \frac{1}{2L} \sum_{t=0}^{T-1} \|\nabla_{\mathrm{bw}} F(S_t)\|_{S_t}^2.$$

Therefore,

$$\frac{1}{T} \sum_{t=0}^{T-1} \|\nabla_{\mathrm{bw}} F(S_t)\|_{S_t}^2 \leq \frac{2L(F(S_0) - F(S_T))}{T} \leq \frac{2L(F(S_0) - F_*)}{T}.$$

$\square$

