# OpenReview forum: "Fréchet Regression on the Bures-Wasserstein Manifold"
_ICLR.cc/2026/Workshop/GRaM — ICLR 2026 Workshop GRaM Poster_

### Official Review · Reviewer_bwBa · 2026-02-18
**This paper studies Fréchet regression with signed weights on the Bures–Wasserstein (BW) manifold of SPD matrices, motivated by the fact that extrapolation in regression can yield negative weights and make the BW Fréchet objective non-coercive, with minimizers that may not exist (illustrated by a concrete failure case). The authors formalize the signed BW objective [ \min_{S\in \mathbb{S}^d_{++}} F(S):=\sum_{k=1}^n \lambda_k, W_2^2(S,\Sigma_k), ] and introduce a Spectral Dominance condition on the positive versus negative mass that guarantees existence of a solution and provides bounds on stationary points. Under this condition, they give a projection-free Riemannian gradient descent method with a sublinear convergence guarantee, and propose a pairwise stochastic reformulation to safely apply SGD even when some weights are negative (with a slightly stronger dominance condition for per-step feasibility). Experiments on temporal ant social networks suggest BW regression better preserves topological signatures (for example, degree-distribution proximity) than Frobenius and Fisher–Rao baselines, and large-scale synthetic diffusion-tensor regression (up to (10^5) tensors) demonstrates rapid and stable optimization plus visually accurate tensor recovery along a helix. Brief summary of my review The paper is a well-motivated and technically clean note on making BW Fréchet regression robust to signed weights via a sufficient existence condition and projection-free optimization, but it would benefit from stronger empirical evaluation and clearer practical guidance on how conservative or checkable the dominance conditions are in typical regression pipelines.**

**Rating:** 7
**Confidence:** 5

**Review:**

Review
Quality and technical soundness
The core technical contribution is the Spectral Dominance sufficient condition (Equation (4)) ensuring well-posedness of the signed BW barycenter problem, addressing the key failure mode where negative weights can push iterates to the boundary or make the infimum unattained. In particular, the condition
[
\sum_{i\in I}\lambda_i^+ \sqrt{\lambda_{\min}(\Sigma_i)}
;>;
\sum_{j\in J}\lambda_j^- \sqrt{\lambda_{\max}(\Sigma_j)}
]
yields existence and spectral bounds on stationary points. The optimization side is also coherent: the paper provides an RGD update with a retraction-like step and a sublinear convergence statement, together with an argument that updates remain in (\mathbb{S}^d_{++}) without explicit projection under the dominance assumptions. The pairwise stochastic reformulation is a sensible way to avoid unsafe SGD steps caused by sampling a single negative-weight term; the stronger per-step condition (6) is clearly stated.
Main technical questions that remain for me:
•	Conservativeness and necessity. The dominance inequalities are sufficient; it is unclear how tight they are or how often they hold in realistic regression (especially in extrapolation, where negative weights can be substantial).
•	Dependence on dimension and conditioning. Even if explicit projection onto (\mathbb{S}^d_{++}) is avoided, computing BW geometric means typically still involves matrix square roots and can be (O(d^3)). The paper makes the projection cost point explicitly, but a more explicit end-to-end complexity discussion would improve credibility for high-dimensional SPD use cases.
Clarity and presentation
For a tiny-paper format, the structure is good: motivation, well-posedness condition, deterministic and stochastic algorithms, then two empirical sections. I did, however, want more “how to use this in regression” clarity: the objective is defined abstractly for signed weights, but typical Fréchet regression weights depend on kernels, local linear fits, or basis expansions, and the paper would be stronger if it explicitly explained how those (\lambda_k) are produced and normalized (and how the dominance condition translates into a practical extrapolation range).
Originality
The signed-weight regime for Wasserstein-type barycenters is a real pain point, and the paper directly targets it by (i) providing a clear sufficient condition for existence and (ii) aligning algorithm design with feasibility (projection-free updates) and scalability (pairwise SGD). The ideas feel more like a solid synthesis plus a clean sufficient-condition argument than a large conceptual leap, but that is acceptable for GRaM, provided the practical impact is well demonstrated.
Significance
If the dominance condition is not overly conservative in practice (or can be enforced by simple regularization), this can be a useful building block for SPD-valued regression in applications like dynamic networks and diffusion tensors, where BW geometry is attractive. The ant network and (10^5)-tensor demonstrations are aligned with this significance claim.

Pros
•	Well-motivated problem formulation: signed weights arise naturally from extrapolation and can break existence.
•	Concrete pathology example showing non-existence without constraints.
•	Clear sufficient condition (Spectral Dominance) yielding existence and bounds on stationary points.
•	Projection-free optimization with a stated sublinear convergence guarantee for RGD.
•	Safe stochastic scaling strategy via pairwise reformulation with an explicit per-step feasibility condition.
•	Empirical evidence in two domains: network regression with topology-oriented metrics and large-scale diffusion tensors with convergence behavior and qualitative tensor fidelity.
Cons
•	Practical enforceability of the dominance conditions is under-discussed. It is not clear how often (4) or the stronger (6) holds for real regression weights, nor how conservative these conditions are.
•	Limited empirical scope: one small real network dataset (one colony, 11 days) plus one synthetic DTI benchmark; there is limited ablation, uncertainty quantification, or comparison to additional SPD baselines (for example, log-Euclidean or affine-invariant Riemannian approaches).
•	Complexity discussion is incomplete for higher-dimensional SPD matrices: projection-free is helpful, but BW geometric mean computations can still be expensive and numerically delicate for ill-conditioned matrices.
•	Regression pipeline details (how (\lambda_k) are computed in typical Fréchet regression and how extrapolation strength maps to feasibility) could be clearer.
Suggestions for improvement
•	Add a short section translating dominance into practice: given a weight construction (kernel regression, local linear), characterize the extrapolation regime that violates (4)/(6), and provide a simple mitigation (for example, regularization toward (\alpha I), adaptive truncation of negative weights, or a certified extrapolation radius).
•	Expand experiments: include at least one higher-dimensional covariance regression task (even moderate (d)), plus additional baselines and an ablation showing failure modes when the condition is violated.
•	Provide more explicit computational details: per-iteration cost and implementation notes for BW geometric means and retractions, plus stability guidance for near-singular inputs.

**Pmlr Suitability:**

Yes

---

### Official Review · Reviewer_wRDg · 2026-02-25
**This work studies  barycenters in the space of positive-definite matrices with the Bures-Wasserstein metric**

**Rating:** 7
**Confidence:** 4

**Review:**

This paper studies Fréchet regression (conditional barycenters) on the Bures–Wasserstein manifold of SPD matrices. The main contributions are: 1) Theorem 2.2 provides a condition for existence of minimizer. 2) projection-free Riemannian GD update and scalable Pairwise Riemannian SGD for large datasets. This is an interesting and valuable contribution and has possible applications to variational inference.

**Pmlr Suitability:**

NA

---

### Meta-Review · Area_Chair_yZnX · 2026-02-26

**Decision:**

Accept

**Metareview:**

This is a neat paper that gives a "spectral dominance" condition, guaranteing the well-poseness of the Frechet regression (in particular for extrapolation). The ant colony experiment is a lovely and original application that nicely illustrates the advantages of the Bures-Wasserstein geometry for preserving network structure. We are happy to accept this tiny paper to GRaM!

One thing that would make the paper even stronger is to explicitly write down the equations for all three metrics compared in the experiments: Bures-Wasserstein, Frobenius, and "Fisher-Rao". On that last point, what the authors call the Fisher-Rao metric is (maybe) more commonly known in the SPD matrix community as the affine-invariant metric. The two do coincide for zero-mean multivariate Gaussians, but this might not be a well-known fact for people coming from the information geometry community, who may be confused about the use of Fisher Rao without probability distributions involved. Writing the metrics explicitly would prevent any confusion.

**Relevance To Proceedings:**

Tiny paper — does not apply

**Relevance To Workshop:**

Yes — suitable for GRaM

---

### Decision · Program_Chairs · 2026-03-02

Accept (Poster)